# Comparison of Japanese Encephalitis Force of Infection in Pigs, Poultry and Dogs in Cambodian Villages

**DOI:** 10.3390/pathogens9090719

**Published:** 2020-09-01

**Authors:** Héléna Ladreyt, Heidi Auerswald, Sothyra Tum, Sreymom Ken, Leangyi Heng, Saraden In, Sokchea Lay, Chakriyouth Top, Sowath Ly, Veasna Duong, Philippe Dussart, Benoit Durand, Véronique Chevalier

**Affiliations:** 1Epidemiology Unit, Laboratory for Animal Health, French Agency for Food, Environmental and Occupational Health and Safety (ANSES), University Paris-Est, 14 rue Pierre et Marie Curie, 94700 Maisons-Alfort, France; helena.ladreyt@anses.fr (H.L.); benoit.durand@anses.fr (B.D.); 2International Center of Research in Agriculture for Development (CIRAD), UMR ASTRE, F-34090 Montpellier, France; 3Virology Unit, Institut Pasteur du Cambodge, Institut Pasteur International Network, 5 Monivong Boulevard, P.O Box. 983, Phnom Penh 12201, Cambodia; hauerswald@pasteur-kh.org (H.A.); ksreymom@pasteur-kh.org (S.K.); hleangyi@pasteur-kh.org (L.H.); insaraden@pasteur-kh.org (S.I.); youthtop@yahoo.com (C.T.); dveasna@pasteur-kh.org (V.D.); pdussart@pasteur-kh.org (P.D.); 4National Animal Health and Production Research Institute, General Directorate for Animal Health and Production, Ministry of Agriculture, Forestry and Fisheries, Phnom Penh 12201, Cambodia; sothyratum@gmail.com; 5Immunology Unit, Institut Pasteur du Cambodge, Institut Pasteur International Network, 5 Monivong Boulevard, P.O Box. 983, Phnom Penh 12201, Cambodia; lsokchea@pasteur-kh.org; 6Epidemiology and Public Health Unit, Institut Pasteur du Cambodge, Institut Pasteur International Network, 5 Monivong Boulevard, PO Box 983, Phnom Penh 12201, Cambodia; lsowath@pasteur-kh.org; 7International Center of Research in Agriculture for Development (CIRAD), UMR ASTRE, Phnom Penh 12201, Cambodia

**Keywords:** Japanese encephalitis virus, multi-host, dog, force of infection, Cambodia

## Abstract

Japanese encephalitis virus (JEV) is the main cause of human viral encephalitis in Asia, with a mortality rate reaching 30%, mostly affecting children. The traditionally described cycle involving wild birds as reservoirs, pigs as amplifying hosts and *Culex* mosquitoes as vectors is questioned, with increasing evidence of a more complex multi-host system involved in areas where densities of pigs are low, such as in Cambodia. In 2018, we examined pigs, chickens, ducks and dogs from Kandal province, Cambodia, for antibody response against JEV by hemagglutination inhibition and virus neutralization assays. Forces of infection (FOI) for flaviviruses and JEV were estimated per species and per unit of body surface area (BSA). JEV seroprevalence reached 31% (95% CI: 23–41%) in pigs, 1% (95% CI: 0.1–3%) in chickens, 12% (95% CI: 7–19%) in ducks and 35% (95% CI: 28–42%) in dogs. Pigs were most likely to be infected (FOI: 0.09 per month), but the FOI was higher in ducks than in pigs for a given BSA (ratio of 0.13). Dogs had a lower FOI than ducks but a higher FOI than chickens (0.01 per month). For a given BSA, dogs were less likely to be infected than pigs (ratio of 1.9). In Cambodia, the virus may be circulating between multiple hosts. Dogs live in close contact with humans, and estimating their exposure to JEV infection could be a relevant indicator of the risk for humans to get infected, which is poorly known due to underdiagnosis. Understanding the JEV cycle and developing tools to quantify the exposure of humans is essential to adapt and support control measures for this vaccine-preventable disease.

## 1. Introduction

Japanese encephalitis virus (JEV) is the major cause of human viral encephalitis in Southeast Asia. JE is also detected in the rest of Asia, from India in the west to China in the north and Japan in the east, as well as in northern Australia. Its annual incidence has been estimated at around 68,000 cases per year, although these figures may be underestimated since the disease occurs mainly in developing countries where access to care and case identification may be limited [1,2,3]. Approximately three quarters of cases concern children and JE remains a substantial public health issue even in areas where human vaccination programs are implemented [1]. The fatality rate can reach 30%, and up to 50% of survivors suffer neurological or psychiatric sequelae [4].

It is commonly accepted that JEV is transmitted between reservoirs (Ardeid birds) or amplification hosts (pigs) through the bites of *Culex* mosquitoes and probably some *Aedes* mosquitoes [5,6,7]. Besides humans and horses, other animals, such as dogs, may be susceptible to JEV infection and are, for now, considered to be dead-end hosts. Indeed, JEV-neutralizing antibodies (nAb) have been found in dogs in Japan [8], but experimentally infected dogs (n = 3) showed very low viremia [9].

However, the circulation of JEV may occur as a consequence of a more complex epidemiological system than usually described. Although pigs occupy a central position, many questions remain about the existence of secondary reservoirs [10]. Indeed, JEV has been shown to circulate in areas where there are few or no pigs [11]. On the other hand, domestic birds have been shown to be exposed to JEV, and experimental infections demonstrated that they are able to develop sufficient viremia to be infectious for mosquitoes, suggesting that poultry could act as a secondary reservoir [12,13,14]. A reservoir can be defined as one or more epidemiologically linked populations in which a pathogen is permanently maintained and from which the infection can be transmitted to a target population [15]. Some authors have studied the concept of such maintenance communities, which may be applied to the circulation of JEV, particularly in Cambodia [16]. A comprehensive approach appears to be essential for understanding the JEV epidemiological cycle, which may be based on distinct maintenance communities according to the area of concern.

In Cambodia, JE is recognized as the most common cause of encephalitis, particularly in children and adolescents, although the exact seroprevalence is not known [17]. Between 2010 and 2013, a survey conducted in two Cambodian hospitals showed that a quarter of acute meningoencephalitis was due to JEV infection [18]. In 2018, more than 45% of childhood encephalitis cases treated in one hospital in Phnom Penh were confirmed to be related to JE (Institut Pasteur du Cambodge personal communication). In Cambodia, most of the field studies on JEV infection and circulation focused on humans, pigs and mosquitoes. They indicated that JEV is endemic and widespread in Cambodia. Regarding pigs, a large serological survey implemented between 2006 and 2007 in farms and slaughterhouses showed that more than 60% of the sampled pigs had been infected with JEV [19]. A longitudinal serosurvey performed in sentinel piglets in peri-urban and rural areas suggest (i) a high circulation of JEV, as the majority of the animals seroconverted within 4 months, and (ii) little seasonal variation in JEV circulation among pigs, despite the existence of a monsoon season in Cambodia (force of infection (FOI) per day, rainy season: 0.032 (sd = 0.006 per day), dry season: 0.046 (sd = 0.008 per day)) [20,21].

Quantifying the relative exposure to JEV of each species, particularly pigs and domestic birds, is a first step to document and study a multi-host transmission system. The probability of a susceptible host to become infected by a given pathogen per time unit can be estimated based on seroprevalence data and can be expressed as forces of infection (FOI). In 2018, we conducted a serological survey in pigs, chickens, ducks and dogs in Kandal Province, Cambodia. Since global human exposure is difficult to estimate due to underdiagnosis, it is indeed questionable to what extent dogs, which live in close proximity to humans, could be a good proxy for human exposure. We used serological data to estimate and compare FOI exerted on individuals of these four species. The relative importance of these hosts in the JEV epidemiological cycle, the existence of a complex multi-host JEV epidemiological system, as well as the potential use of dog as sentinels are discussed.

## 2. Results

### 2.1. Description of Samples and Seroprevalence

Out of the 666 sampled animals, seven pigs and eight chickens were excluded because their age was unknown, and three pigs were excluded because there was insufficient serum to undergo a foci reduction neutralization test (FRNT). Overall, 648 animal sera were tested by hemagglutination inhibition assay (HIA) (112 pigs, 128 ducks, 220 chickens and 188 dogs) and 134 samples (21%) were positive for flavivirus antibodies. We observed a flavivirus seroprevalence of 31% for pigs (95% CI: 23–41%), 2% for chickens (95% CI: 1–5%), 15% for ducks (95% CI: 9–22%), and 40% for dogs (95% CI: 33–48%). Among these HIA-positive sera, 125 revealed neutralizing antibodies (nAb) detectable by FRNT. Based on the assumptions of (i) a low or moderate seroprevalence for poultry and dogs and (ii) that secondary host animals have in general lower antibody titers, we used the less stringent FRNT_50_ to increase the sensitivity of our survey. The JEV seroprevalence based on the FRNT_50_ results (i.e., the proportion of JEV-positive and JEV and dengue virus (DENV) double-positive animals) was 31% for pigs (95% CI: 23–41%), 1% for chickens (95% CI: 0.1–3%), 12% for ducks (95% CI: 7–19%), and 35% for dogs (95% CI: 28–42%) (Table 1).

In pigs, JEV seroprevalence did not significantly differ between farms and backyards with 26% (95% CI: 14–40%) and 35% (95% CI: 24–48%), respectively (Fisher’s test *p* = 0.3). In chickens, 1% of the animals were JEV-positive in farms (95% CI: 0–6%) and 1% in backyards (95% CI: 0–4%) (Fisher’s test *p* = 1). In ducks, JEV seroprevalence differed significantly between farms and backyards: 41% in farms (95% CI: 25–59%) against 1% in backyards (95% CI: 0–6%) (Fisher’s test, *p* < 10^−4^), although ducks were older in farms than in backyards (median age of 6 and 2 months, respectively). Indeed, JEV seroprevalence significantly increased with the age group (Fisher’s test, pigs: *p* = 0.0005; ducks: *p* < 10^−4^; dogs: *p* < 10^−4^), except for chickens (Fisher’s test, *p* = 0.17).

### 2.2. Multivariate Analysis

Based on HIA and FRNT results, we analyzed the risk of flavivirus and JEV infection, respectively (Table 2). Animals from Angk Snuol and S’ang districts (n = 35) were excluded from the dataset as no positive sample were found. Compared to pigs, JEV seropositivity risk was significantly lower in chickens (OR = 0.03, 95% CI: 0.01–0.12), and in dogs (OR = 0.38, 95% CI: 0.18–0.79). Oppositely, we detected no significant difference of seropositivity risk between ducks and pigs. After controlling the species effect, age increased the JEV seropositivity risk, with an odd ratio of 1.14 (95% CI: 1.06–1.23) for a difference of 6 months. Similar results were obtained for the risk of flavivirus seropositivity, except that the difference between dogs and pigs was not significant (Table 2).

### 2.3. Estimation of the Force of Infection (FOI) for Flaviviruses and JEV in the Sampled Animals

The FOI estimates for JEV were 0.09 per month for pigs, 0.03 per month for ducks, 0.01 per month for dogs, and 0.003 per month for chickens (Table 3); a pig was three times more likely to get JEV infected than a duck, and a duck was ten times more likely to be JEV infected than a chicken. As the confidence intervals of estimated FOI do not overlap, the probability of JEV infection was significantly higher in pigs than in ducks, higher in ducks than in dogs, and also higher in dogs than in chickens. Same results were found for flavivirus FOI. In order to rule out a “district” effect on the estimated FOIs, we estimated the FOIs for Khsach Kandal only, as it was the only district where all four species were sampled (Appendix A). Estimated FOIs were similar to those estimated for all districts, except for chickens, which were all found negative in Khsach Kandal.

### 2.4. Force of Infection per Body Surface Area (BSA)

In the present study, the median weight was 90 kg for pigs (median age of 4 months), 0.8 kg for chickens and ducks (median age of 2 months), and 15 kg for dogs (median age of 25 months). The ratios of FOI per unit of body surface area (BSA) was 0.13 between pigs and ducks, indicating that, for a given BSA, ducks are more likely to get JEV infected than pigs. In contrast, the ratios of FOI per unit of BSA was 1.3 between pigs and chickens, and 1.9 between pigs and dogs, suggesting that both chickens and dogs are less likely to get JEV infected than pigs for a given BSA.

### 2.5. Comparison of FOI for Poultry in 2016 and 2018

The estimated FOI for flaviviruses was 0.05 per month (95% CI: 0.04–0.07) for ducks and 0.03 per month in chickens (95% CI: 0.02–0.03) sampled in Kandal province in 2016.

## 3. Discussion

JEV is known to be circulating in pigs in almost all Southeast Asian countries [22]. Ducks have been shown to be exposed to JEV (JEV nAb detection) in Cambodia and India, and chickens in Cambodia and Singapore [14,23,24]. Finally, JEV nAb have been detected in dogs in Singapore and Japan [8,24]. Our survey, conducted in 112 pigs, 128 ducks, 220 chickens and 188 dogs from five districts of Kandal province, Cambodia, confirms that Japanese encephalitis circulates in this province and that the four tested species are exposed to JEV to different extents. The observed JEV seroprevalence was similar for dogs (35%) and pigs (31%), whereas the multivariate analysis showed that the seropositivity risk was higher for ducks and pigs. For all four species, the seropositivity risk increased with age.

Overall, the seroprevalence calculated from our samples was lower than that reported in other studies, especially for chickens. This is probably due to the age of the sampled animals; with the exception of dogs, most of the animals we sampled were very young, so the possible exposure period was short. Depending on the studied region (Nepal, Bali, Vietnam and Cambodia), flavivirus seroprevalence ranged from 32% to 65% in pigs, from 24% to 37% in chickens, and from 31% to 40% in ducks [14,19,25,26,27]. In a former JEV seroprevalence study on poultry in Kandal province in 2016 [14], a statistical model of seroprevalence by age predicted a seroprevalence of 5–15% in 3-month-old chickens. We observed a lower flavivirus seroprevalence in chickens with a median age of 3 months (1%, 95% CI: 0.5–4.6%). In the same study, the seroprevalence in 3-month-old ducks was predicted to be 15–35%, which is consistent with the flavivirus seroprevalence we found in ducks (12%, 95% CI: 9.2–22.2%).

The JEV seroprevalence we determined using FRNT could not be directly compared with the above-mentioned cross-sectional studies, as the serological analysis methods differed. In India, authors using microtiter virus neutralization test all the HIA-positive samples and reported a JEV seroprevalence in ducks of 19.1% [23]. A study led in Singapore (using a more stringent interpretation threshold FRNT_80_) showed high seroprevalence levels with 60% of the sampled chickens (n = 20) and 39.6% of the sampled dogs revealing JEV nAb [24].

As expected, pigs had the highest monthly probability of acquiring JEV infection (JEV FOI = 0.09 per month). Interestingly, ducks followed with a JEV FOI of 0.03 per month, then dogs (JEV FOI of 0.01 per month). Chickens were the least likely to be infected, with an estimated JEV FOI of 0.003 per month. Regarding the monthly probability of acquiring flavivirus infection, comparing the flavivirus FOI values of our study (2018) with the ones estimated from the survey led in Kandal in 2016 [14] revealed similar probability of flavivirus infection per month in ducks (0.05 per month in 2016 (95% CI: 0.04–0.07), 0.04 per months in 2018 (95% CI: 0.03–0.06)). This latter result supports the fact that, for ducks, there is little seasonality and little inter-annual variation of the intensity of JE transmission in this province. The FOI for flaviviruses in chickens was higher in 2016 than in 2018 (0.03 per month in 2016 (95% CI: 0.02–0.03), 0.007 per month in 2018 (95% CI: 0.003–0.01)), a difference for which we could not find any plausible explanation given the available data.

The JEV FOI exerted on an individual is related to its exposure to the bites of a competent vector. It depends on the trophic preference of the vector, and on a combination of both host attractiveness and host body surface area (BSA). Reports on host feeding patterns of JEV vectors showed that *Culex* species are often opportunistic. In China and Senegal, *Cx. tritaeniorhynchus*, a major vector of JEV, was shown to feed on multiple hosts, such as as swine, poultry, dog, cattle and human, in different proportions [28,29]. In India, some studies confirmed that host availability plays an important role in the feeding behavior of *Cx. tritaeniorhynchus* [30,31]. Concerning FOI per host BSA unit, the ratio we calculated between pigs and ducks was less than one, suggesting that, for a given BSA, ducks were more likely to get infected than pigs. Since ducks develop viremia high enough to infect back mosquitoes [13,32], further studies should be implemented to better understand their role in JEV epidemiological cycle and decipher the relative contribution of pigs and ducks in JEV circulation. Oppositely, the ratios of FOI per BSA unit for chickens and dogs in relation to pigs were greater than 1, suggesting that pigs are more exposed than chickens and dogs for a given BSA. Under experimental conditions, infected chickens seem to develop a lower viremia than ducks, suggesting that they would be a less efficient reservoir for JEV transmission [13]. However, density estimates of the different species in the study area of Kandal suggest that there are about ten times more chickens than ducks (unpublished data, field observation: about 10,000 chickens and 900 ducks in an average village of Kandal province). In addition, chickens are slaughtered when they are around 6 months old, unlike ducks, which are reared for meat until they are over 2 years old. Thus, the abundance and high population renewal (hence frequent introduction of naïve hosts) of chickens could compensate for the lower FOI and make them an important link in the JEV epidemiological cycle.

Our study confirms the central role of pigs for maintaining JEV transmission. In countries where pigs are housed far away from human housing, or in countries with little to no pig farming, the use of pigs as sentinels for JEV circulation may be limited. Therefore, other domestic animals that are housed in close proximity to humans should be investigated as a suitable sentinel. Previous studies showed that dogs could be good sentinels for West Nile virus [33,34], Lyme disease [35] and other arthropod-borne diseases [36]. In addition, exposure of dogs to JEV has already been observed in Singapore and Japan [8,24]. The present study is the first evidence of exposure of dogs to JEV in Cambodia. Experimental JEV infections in dogs showed that they developed a long-lasting immunity but no clinical signs and a low viremia [9]. In addition, it is much easier to obtain samples from dogs than from pigs or poultry, as owners are much less reluctant to participate in studies because dogs have a lower economic value. In Cambodia, dogs are numerous with an estimated dog/human ratio as high as 1:3 in some areas (V. Chevalier, personal communication) and live in close proximity to humans even when they are free roaming. Dogs could therefore be a good proxy for human exposure to JEV. Even though JE is known to be endemic and widespread in Cambodia, the spatial and temporal distribution of the disease is not well known, as studies are focused on the capital, Phnom Penh, and neighboring Kandal province. Therefore, using dogs as sentinels would help investigating the high circulation level of the virus and identifying the main risk areas. This highlights the need to invest more in JEV surveillance, research and vaccination in Cambodia. Sentinel dogs may also be a relevant choice for establishing surveillance in a country where JEV circulation follows an epidemic pattern, or even serve as an early warning system for JEV introduction in a country free of the disease.

As others have done before [12,37], we challenge the dogma that limits the JEV transmission cycle to the involvement of pigs, vectors and wild birds. Our study demonstrates the circulation of JEV in a broader, multi-host system involving several domestic animal species. Haydon et al. (2002) defined a reservoir as one or more epidemiologically connected populations or environments in which the pathogen can be permanently maintained and from which infection is transmitted to a defined target population [15]. In our case, host diversity may explain why JEV circulates intensively even in an ecosystem where the density of its main host, pigs, is relatively low. The maintenance of JE could be facilitated not by a single reservoir species but by a maintenance community composed of pigs, ducks, chickens and vectors. Roberts and Heesterbeek (2020) pointed out that the reservoir capability of a (group of) species may be subject to changes in the ecosystem [16]. Depending on environmental conditions, a given species may be more or less involved in the epidemiological cycle. Our next step will be to develop a mathematical model of JEV circulation in Kandal province to decipher and quantify the respective roles of the different host species in JEV transmission. Additional, transmission studies should investigate the vector competence of local *Culex* species and the potential involvement of poultry and dogs in the JEV transmission cycle. This may help to better understand the dynamics of JEV circulation in the maintenance community, under various contexts, which is necessary to better monitor, estimate the risk of infection in humans, and thus reduce the impact of JE on human health, especially in children.

## 4. Materials and Methods

### 4.1. Study Area

Kandal is a large province (3179 km^2^), encompassing 11 districts and surrounding Phnom Penh, the capital of Cambodia. The province consists of a typical plain wet area, dominated by cultivated fields and a mosaic of rural and peri-urban areas. The climate is warm and humid, with monsoon season between May and October. In this area, pigs and poultry are raised either in backyards (traditional farming system where animals are raised often in low densities within the confines of the family home) or in farms (so-called modern breeding system in specific installations).

The region is characterized by a low pig density. Even though pig farming in Cambodia is undergoing profound changes due to the arrival of industrial farming, driving meat price down, traditional pig raising in backyards still predominates. Pig production is concentrated in the provinces around Phnom Penh, including Kandal. In 2004, FAO estimated a pig density in Kandal province of about 50 animals per square km [38,39]. As a comparison, the density of pigs reared in Bretagne region, France, is more than 250 per square km [40]. The districts and villages were selected according to their accessibility (breeder’s willingness and access to farms and backyards).

### 4.2. Dog Samples

Dog sera were acquired during the research project “Man’s best friend: A cross border transdisciplinary One Health approach to rabies control in dogs in Southeast Asia” that started in March 2018. A systematic and comprehensive door-to-door visit was conducted in 10 villages of Khsach Kandal district in Kandal province. Each encountered dog was individually identified and data related to the age, sex, mode of confinement, and relations with the family were recorded. Each dog was vaccinated against rabies. Among 2200 recorded dogs, 800 were blood sampled. Overall, 188 samples were included based on a sufficient serum volume for rabies and JEV serology.

### 4.3. Samples of Domestic Pigs, Chickens, and Ducks

Between October and December 2018, 122 pigs, 228 chickens and 128 ducks from farms, backyards, and slaughterhouses of 21 villages of Kandal province were sampled (Figure 1). Farms and backyards were recruited based on the willingness of owners. Sampled animals were randomly selected among those over one month of age for poultry, and over two months of age for pigs, to avoid interfering with maternal immunity [20,41]. Since our objective was to study the forces of infection and since we know that seroprevalence is high in pigs and increases with age, we focused on backyard and farm pigs, which are slaughtered before the age of 6 to 7 months. All the animals sampled were selected to maximize age diversity for estimation of the forces of infection. Sera were collected in 35 backyards, seven farms (3 farms for pigs, 2 for ducks and 2 for chickens) and five slaughterhouses (4 for pigs and 1 for chickens), located in the district where the dogs had been sampled (Khsach Kandal) and in two neighboring districts (Kean Svay and Koh Thum). One hundred animals (pigs and chickens) had to be sampled from the slaughterhouse, as breeders were sometimes reluctant to give access to their animals since these ones are often their only source of income. However, these animals had just been delivered (usually the same morning), and the ages and districts of origin were precisely known by the slaughterhouse managers. They originated from the districts of the slaughterhouse, although some of the chickens came from three neighboring districts of Kandal province (Angk Snuol and S’ang, n = 35) (Figure 1).

The age of each sampled animal was recorded, based on farmer’s declarations or slaughterhouse staff. Pigs ranged in age from 2 to 6 months, chickens from 1 to 24 months and ducks from 1 to 15 months. Blood samples were taken from the jugular vein in pigs and the ulnar or metatarsal vein in poultry. Specimens were collected on dry blood tubes and kept at 4 °C after sampling, and transported to the Institut Pasteur du Cambodge (IPC), where the serum was separated immediately by centrifugation and was frozen at −80 °C until further analysis.

### 4.4. Ethic Statement

All samples used in this study were collected following the World Animal Health Organization (OIE) guiding principles on animal welfare included in the OIE Terrestrial Animal Health Code [42]. All sampling campaigns were implemented under the supervision of the Cambodian General Direction of Animal Production and Health (GDAPH), and local veterinary services.

### 4.5. Laboratory analysis

All samples were analyzed by the Virology unit of the IPC. They were first tested by hemagglutination inhibition assay (HIA). Positive samples were subsequently tested for neutralizing antibodies (nAb) against JEV and dengue virus serotype 3 (DENV-3) by foci reduction neutralization test (FRNT) if a sufficient volume of sera remained. The strategy to test for both JEV and DENV was chosen because DENV is known to be widespread in the region [43,44] and to induce cross-reactions with anti-JEV antibodies [45].

#### 4.5.1. Cells and Viruses

Simian Vero cells (ATCC CCL-81) were used for the detection of neutralizing antibodies via FRNT. These were cultivated in Dulbecco’s modified Eagle medium (DMEM; Sigma-Aldrich, Steinheim, Germany) supplemented with 10% fetal bovine serum (FBS; Gibco, Gaithersburg, MD, USA) and 100 U/mL penicillin-streptomycin (Gibco) at 37 °C and 5% CO_2_ atmosphere. All viruses were grown in C6/36 *Aedes albopictus* cells. These mosquito cells were cultivated in Leibovitz-15 medium (Sigma-Aldrich) supplemented with 10% FBS, 1% L-glutamine (Gibco), 10% tryptose-phosphate (Gibco) and 100 U/mL penicillin-streptomycin at 28 °C. JEV strain Nakayama (Genbank EF571853) and Dengue 3 (DENV-3) strain H87 (Genbank M93130) were used for HIA and FRNT.

#### 4.5.2. Hemagglutination Inhibition Assay (HIA)

The presence of antibodies in the serum samples was analyzed with HIA using antigen originated from the above-mentioned JEV and DENV-3 strains. The assay followed the protocol previously described [46] adapted to 96 well microtiter plates.

#### 4.5.3. Foci Reduction Neutralization Test (FRNT)

The FRNT determined the level of neutralizing antibodies and was performed as described previously [47], modified by using Vero cells. The titer of neutralizing antibodies (nAb) was expressed as the reciprocal serum dilution that induces 50% reduction of infection visualized as foci (FRNT_50_) compared to the controls (flavivirus-negative serum and virus control) and was calculated via log probit regression analysis (SPSS for Windows, Version 16.0, SPSS Inc., Chicago, IL, USA). FRNT_50_ titers below 10 were considered negative. FRNT results were classified in four outcomes depending on the neutralizing antibody titer against JEV and DENV: (i) “JEV positive” if only JEV nAb were detected or if JEV nAb titer was four times higher than DENV nAb titer; (ii) “DENV positive” if only DENV nAb were detected or if DENV nAb titer was four times higher than JEV nAb titer; (iii) “JEV and DENV double-positive” if both JEV and DENV nAb were detected with similar nAb titers (less than a factor 4 difference); (iv) “Negative” when no nAb were detected. In the latter case, the conclusion was “Flavivirus positive”, as the samples were all HIA positive.

### 4.6. Statistical Analysis

Samples that could not be tested in FRNT after a positive HIA result (due to an insufficient volume of serum) and those for which the age of the animal was not available were discarded from the analysis.

We used logistic mixed models to analyze the relationship between the seropositivity risk and two variables: the species (pig, duck, chicken or dog) and the age (in months). The effect of possible geographic variations of seropositivity risk was controlled by adding the district of origin as a random effect (to allow model convergence, samples from districts where no positive animal had been observed were excluded from the datasets used). Two logistic mixed models were fitted: one for HIA results and the other for FRNT results. We considered that the animal was seropositive to JEV when the result of the FRNT test was “JEV positive” or “JEV and DENV double-positive”, since both cases indicate a previous JEV infection in the sampled animal.

The force of infection (FOI) is the instantaneous probability of a susceptible host to become infected. Because of the relatively small surface of the survey area, and low seasonality of JEV circulation in Kandal province, the FOI was assumed constant [20,21]. Under this assumption, the force of infection for flaviviruses (based on HIA results) and JEV (based on FRNT results) can be estimated by maximizing the joint likelihood of age-specific seroprevalence data. The probability for an animal to be seronegative (1) or seropositive (2) was respectively expressed as:(1)$$P\left(neg\right)=\;e^{-\lambda \alpha }$$
(2)$$P\left(pos\right)=1-e^{-\lambda \alpha }$$
With *α* the age in months, and *λ* the FOI by month.

The joint likelihood of the observed data (i.e., the product of the above probabilities on all animals included in the study) was then maximized to obtain the estimate of the FOI. This model was successively fitted for each of the four studied species. The variance-covariance matrix, obtained by inverting the Hessian matrix, allowed for computing the confidence intervals. All analyses were done with R software version 3.6.0 (R Foundation for Statistical Computing, Vienna, Austria) [48].

Using the same methodology, we estimated FOI for flaviviruses using a serological dataset of a poultry survey implemented in the same province in March 2016 [14] (Appendix A).

### 4.7. Force of Infection per Unit of Body Surface Calculation

The JEV FOI exerted on an animal of a given host species is related to its exposure to the bites of a competent vector. This exposure depends on a combination of both host body surface area (BSA) and attractiveness and on the trophic preference of the vector. A rough estimate of the FOI per unit of host body surface area (BSA) can be computed using the allometric relationship between mass (M) and BSA: BSA~M^2/3^ (where ~ denotes the proportionality relationship) [49]. This formula was used to calculate the FOI ratio per unit of BSA between each pair of species.

## Figures and Tables

**Figure 1 pathogens-09-00719-f001:**
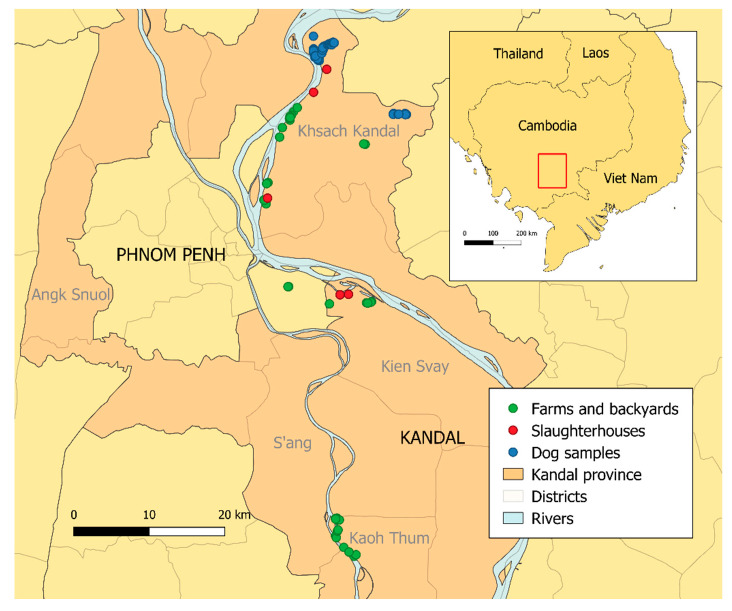
Locations of the sampling sites.

**Table 1 pathogens-09-00719-t001:** Results of serological tests for flavivirus (measured by hemagglutination inhibition assay (HIA)) and Japanese encephalitis virus (JEV) (measured by foci reduction neutralization test (FRNT)) seroprevalence rates.

		HIA (Pos/Tested)	FRNT (Pos/Tested)			Seroprevalence Rates	
			JEV ^a^	DENV ^b^	JEV and DENV ^c^	Flavivirus (HIA) ^d^	JEV (FRNT) ^e^
**Species**							
	**Pigs**	35/112	34/35	0/35	1/35	0.31	0.31
	**Chickens**	4/220	0/4	2/4	2/4	0.02	0.01
	**Ducks**	19/128	2/19	2/19	13/19	0.15	0.12
	**Dogs**	76/188	63/76	4/76	2/76	0.40	0.35
**Districts of origin**							
	**Khsach Kandal**	112/335	84/112	6/112	15/112	0.33	0.30
	**Kean Svay**	18/110	14/18	2/18	2/18	0.16	0.15
	**Koh Thum**	4/168	1/4	0/4	1/4	0.02	0.01
	**Angk Snuol**	0/30	nd	nd	nd	0	nd
	**S’ang**	0/5	nd	nd	nd	0	nd
**Age groups ***							
**Pigs**	**2** **–** **4 months**	9/58	9/9	0/9	0/9	0.16	0.16
	**>4** **–** **5 months**	12/28	11/12	0/12	1/12	0.43	0.43
	**>5** **–** **6 months**	14/26	14/14	0/14	0/14	0.54	0.54
**Chickens**	**1** **–** **2 months**	0/129	nd	nd	nd	0	nd
	**>2** **–** **3 months**	3/59	0/3	2/3	1/3	0.05	0.02
	**>3** **–** **24 months**	1/32	0/1	0/1	1/1	0.03	0.03
**Ducks**	**1** **–** **2 months**	2/65	1/2	0/2	0/2	0.03	0.02
	**>2** **–** **5.5 months**	6/40	1/6	0/6	4/6	0.15	0.13
	**>5.5** **–** **15 months**	11/23	0/11	2/11	9/11	0.48	0.39
**Dogs**	**1.5** **–** **24 months**	30/103	21/30	3/30	1/30	0.29	0.21
	**>24** **–** **48 months**	26/45	25/26	0/26	0/26	0.58	0.56
	**>48** **–** **108 months**	20/40	17/20	1/20	1/20	0.50	0.45

pos: positives; JEV: Japanese encephalitis virus; DENV: Dengue virus (DENV-3 explicitly). * For all species, age groups were determined based on the quartiles of their age distribution; ^a^ Only JEV nAb detected or JEV nAb titer four times higher than DENV nAb titer; ^b^ Only DENV nAb detected or DENV nAb titer four times higher than JEV nAb titer; ^c^ Both JEV and DENV nAb detected with similar nAb titer (less than a factor four difference); ^d^ HIA positives/HIA tested; ^e^ (JEV positives + JEV and DENV double positives)/HIA tested; nd: not done.

**Table 2 pathogens-09-00719-t002:** Results of the multivariate model for flavivirus (measured by HIA) and JEV (measured by FRNT) seroprevalence in pigs, poultry and dogs of Kandal province, Cambodia.

		Flavivirus Seroprevalence (HIA)		JEV Seroprevalence (FRNT)	
Variable	Value	OR (95% CI)	*p*-Value	OR (95% CI)	*p*-Value
**Species**	Pigs	ref.	-	ref.	-
	Chickens	0.07 (0.02–0.19)	<0.0001	0.03 (0.01–0.12)	<0.0001
	Ducks	0.69 (0.31–1.49)	0.34	0.50 (0.22–1.11)	0.09
	Dogs	0.62 (0.31–1.25)	0.18	0.38 (0.18–0.79)	0.009
**Age**	*quantitative*	1.12 * (1.04–1.2)	0.003	1.14 * (1.06–1.23)	0.0004

* Odds Ratios (OR) correspond to an age difference of 6 months.

**Table 3 pathogens-09-00719-t003:** Estimated forces of infection (FOI) per month for flaviviruses and JEV by species.

	Flavivirus FOI (HIA)		JEV FOI (FRNT)	
	FOI	95% CI	FOI	95% CI
**Pigs**	0.09	(0.07–0.11)	0.09	(0.07–0.11)
**Ducks**	0.04	(0.03–0.06)	0.03	(0.02–0.04)
**Dogs**	0.02	(0.01–0.02)	0.01	(0.01–0.02)
**Chickens**	0.007	(0.003–0.014)	0.003	(0.001–0.009)

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
