# Peer review of "Comparison of Japanese Encephalitis Force of Infection in Pigs, Poultry and Dogs in Cambodian Villages"

_pathogens, 2020, doi:10.3390/pathogens9090719_

Round 1

Reviewer 1 Report

This is a well-written manuscript exploring the possibility of secondary reservoirs for JEV. This type of field research is not easily done and I applaud the authors on their efforts. As expected, the seroprevalence and FOI were highest in pigs. It was interesting that dogs were starkly differentially infected between backyards and farms. The only minor omission in the methodology would be a sample size calculation for the various reservoirs that were sampled to ensure that enough animals were sampled to provide sufficient power to these conclusions - the confidence intervals were quite wide for seroprevalence rates in Table 2. However, given the difficulty in sampling these animals and convenience sampling used from rabies studies, I still think that the study is robust.

Author Response

Thank you very much for your positive comments.

In terms of sample size, we had initially planned to collect 50 animals per species from 10 villages (excluding dogs), which would have allowed us to estimate a prevalence of about 30% with a relative precision of +/-12.5% and a confidence level of 95%. Indeed, previous poultry surveys in Cambodia have reported prevalences of about 40% and 24% for ducks and chickens respectively (Auerswald et al, 2020).

Concerning pigs, the large cross-sectional survey conducted in 2008 in Cambodia reported a seroprevalence of about 65%, but the animals were largely collected at the slaughterhouse, and were therefore older (Duong et al, 2011). Since our objective was to study the forces of infection and since we know that seroprevalence is high in pigs and increases with age, we focused on backyard and farm pigs, which are slaughtered before the age of 6 to 7 months (this sentence has been added to the manuscript, line 312). Therefore, it was reasonable to aim for a seroprevalence lower than 65%, given the young age of the animals we were going to collect.

However, field constraints led us to reduce the number of animals collected and to increase the number of villages visited. Several farmers refused to let us sample their pigs for fear of losing animals, because of the great breeding difficulties they encounter (cost of production higher than the cost of meat) but also because of the circulation of ASF in China, reinforcing their mistrust for sanitary reasons. We lost some time trying to find solutions and the funding did not allow us to carry out more missions in order to reach a larger sample size for the 3 species.  

The purpose of the study was to compare infection forces. We did not estimate the statistical power of the study design for FOI estimation, but the confidence intervals derived from the catalytic model estimating the FOI appeared suitable. We indeed tried to maximize the age diversity of the animals we sampled (the sentence “The animals sampled were selected to maximize age diversity for estimation of the forces of infection” has been added line 314).

We hope these answers are satisfactory to you, and we thank you again for your comments and your prompt review.

Reviewer 2 Report

Reviewer comments:

The manuscript by Ladrey and colleagues study quantifying the relative exposure to JEV of each species, to understand multi-host transmission system with title “Comparison of Japanese encephalitis force of infection in pigs, poultry and dogs in Cambodian villages”. The authors are trying to estimate the probability of a susceptible host to become infected by a given pathogen per time unit can be estimated based on seroprevalence data and presenting as forces of infection (FOI). By this method, authors found out JEV seroprevalence reached 31% in pigs, 1% in chickens, 12% in ducks and 35% in dogs.

There are a few minor concerns to the experiments the authors should consider in order to strengthen the conclusions:

  1. Authors have mentioned in discussion that JEV is known to be circulating in pigs, ducks and Chickens in almost all Southeast Asian countries, in Cambodia, India, and Singapore. Also, mentioned that this particular study is from Kandal province, Cambodia. Since these were from domestic animals and birds living closely with humans, what is the JEV seroprevalence in humans living in that area?
  2. Since this virus can also transmit between hosts (Humans-to-domestic animals such as pigs, ducks and dogs), how do you track the virus transmissions?   

In conclusion, I believe this article has drawn important conclusions from this study has and helps in understanding the JEV cycle and developing tools to quantify exposure of humans is essential to adapt and support control measures for this vaccine-preventable disease. Suitable for the publication in pathogens journal with minor revision.

Author Response

Answer to the Comments and Suggestions for Authors

Reviewer comments:

The manuscript by Ladrey and colleagues study quantifying the relative exposure to JEV of each species, to understand multi-host transmission system with title “Comparison of Japanese encephalitis force of infection in pigs, poultry and dogs in Cambodian villages”. The authors are trying to estimate the probability of a susceptible host to become infected by a given pathogen per time unit can be estimated based on seroprevalence data and presenting as forces of infection (FOI). By this method, authors found out JEV seroprevalence reached 31% in pigs, 1% in chickens, 12% in ducks and 35% in dogs.

There are a few minor concerns to the experiments the authors should consider in order to strengthen the conclusions:

  1. Authors have mentioned in discussion that JEV is known to be circulating in pigs, ducks and Chickens in almost all Southeast Asian countries, in Cambodia, India, and Singapore. Also, mentioned that this particular study is from Kandal province, Cambodia. Since these were from domestic animals and birds living closely with humans, what is the JEV seroprevalence in humans living in that area?

Thank you very much for your comments and questions.

Information on human exposure to JEV in Cambodia comes from surveys conducted in hospitals in Phnom Penh and Siem Reap, and therefore not in Kandal, although it can be assumed that some of the patients in Phnom Penh live in Kandal province. These surveys do not make it possible to specify the prevalence in the population because they are carried out only on patients in these hospitals suffering from encephalitis. The seroprevalence of JEV in humans in Kandal, as in the whole of Cambodia, is unknown mainly because of a high level of under-diagnosis. Moreover, to our knowledge, no serological survey has been conducted in humans in Cambodia. This is why we wanted to focus on dogs, which could then be a good proxy for human exposure to JEV. We added a precision to the sentence in the abstract, line 44: “Dogs are living close to humans and estimating their exposure to JEV infection could be a relevant indicator of the risk for humans to get infected, which is poorly known due to under-diagnosis.”; and in the introduction, line 78: “In Cambodia, JE is recognized as the most common cause of encephalitis, particularly in children and adolescents, although the exact seroprevalence is not known”.

  1. Since this virus can also transmit between hosts (Humans-to-domestic animals such as pigs, ducks and dogs), how do you track the virus transmissions?   

Thank you for your question.

Indeed, we did not attempt to track transmission because we assumed, based on previous studies, that JEV is circulating in the region and that it can indeed be transmitted between reservoirs/secondary reservoirs or amplifying hosts (pigs and poultry) by mosquito bites. Human and dogs are dead-end hosts that cannot transmit the virus to mosquitoes. The aim of the study was to provide information on the relative exposure of each host and to estimate the forces of infection in order to help understand the epidemiological cycle of JEV.

To track transmission between hosts, it would be necessary to isolate the virus on each host, including mosquitoes (which is rarely successful because the viremia is usually very short, and the infection rate of mosquitoes is low), and use genetic methods such as whole genome sequencing to demonstrate that the same virus circulates between hosts in a particular pattern. This has been done in few studies that suggested the implication of pigs in JEV cycle, as authors found the same virus genotype in pigs, humans and mosquitoes.

In conclusion, I believe this article has drawn important conclusions from this study has and helps in understanding the JEV cycle and developing tools to quantify exposure of humans is essential to adapt and support control measures for this vaccine-preventable disease. Suitable for the publication in pathogens journal with minor revision.

Thank you, we hope these answers are satisfactory to you, and we thank you again for your comments and for your prompt review.